# The Antibiotic Fosfomycin Mimics the Effects of the Intermediate Metabolites Phosphoenolpyruvate and Glyceraldehyde-3-Phosphate on the *Stenotrophomonas maltophilia* Transcriptome

**DOI:** 10.3390/ijms23010159

**Published:** 2021-12-23

**Authors:** Teresa Gil-Gil, Luz Edith Ochoa-Sánchez, José Luis Martínez

**Affiliations:** 1Centro Nacional de Biotecnología, CSIC, Darwin 3, 28049 Madrid, Spain; tgil@cnb.csic.es (T.G.-G.); luzedith.os@gmail.com (L.E.O.-S.); 2Programa de Doctorado en Biociencias Moleculares, Universidad Autónoma de Madrid, 28049 Madrid, Spain

**Keywords:** *Stenotrophomonas maltophilia*, fosfomycin resistance, transcriptomics, phosphoenolpyruvate, glyceraldehyde-3-phosphate, adjuvant

## Abstract

*Stenotrophomonas maltophilia* is an opportunistic pathogen with an environmental origin, which presents a characteristically low susceptibility to antibiotics and is capable of acquiring increased levels of resistance to antimicrobials. Among these, fosfomycin resistance seems particularly intriguing; resistance to this antibiotic is generally due to the activity of fosfomycin-inactivating enzymes, or to defects in the expression or the activity of fosfomycin transporters. In contrast, we previously described that the cause of fosfomycin resistance in *S. maltophilia* was the inactivation of enzymes belonging to its central carbon metabolism. To go one step further, here we studied the effects of fosfomycin on the transcriptome of *S. maltophilia* compared to those of phosphoenolpyruvate—its structural homolog—and glyceraldehyde-3-phosphate—an intermediate metabolite of the mutated route in fosfomycin-resistant mutants. Our results show that transcriptomic changes present a large degree of overlap, including the activation of the cell-wall-stress stimulon. These results indicate that fosfomycin activity and resistance are interlinked with bacterial metabolism. Furthermore, we found that the studied compounds inhibit the expression of the *smeYZ* efflux pump, which confers intrinsic resistance to aminoglycosides. This is the first description of efflux pump inhibitors that can be used as antibiotic adjuvants to counteract antibiotic resistance in *S. maltophilia*.

## 1. Introduction

Exposure to antibiotics produces global changes in the bacterial transcriptome, in a concentration-dependent manner, including changes in the expression of genes encoding determinants that allow protection against the antibiotic—such as efflux pumps or antibiotic-inactivating enzymes—and of genes not directly involved in this adaptive response [1,2,3]. It has been shown that at low concentrations, the expression of several genes that are unrelated to the bacterial stress-response pathways may change, suggesting that antibiotics may present a hormetic effect [4,5], signaling beneficial molecules at low concentrations and inhibitors at higher ones [2]. In agreement with this concept, when antibiotic concentrations increase, transcriptional changes related to stress responses are found. The function of stress-response genes is to maintain cellular homeostasis, compensating for alterations suffered as a result of different stressors, including antibiotics. In addition to determinants of antibiotic resistance (see above), stress-response networks include genes that mediate general stress responses, for instance, heat-shock proteins that enable cell survival at high temperatures and in other stressful situations [6]. Finally, when concentrations of antibiotics further increase, transcriptional changes likely reflect the loss of bacterial viability [3,7].

*Stenotrophomonas maltophilia* is an opportunistic pathogen of great public health concern, due to its reduced susceptibility to antibiotics; it is an aerobic, non-fermentative, Gram-negative bacterium with environmental origin [8,9], and is commonly associated with respiratory infections [10]. A remarkable fact is that the treatment of infections caused by *S. maltophilia* is frequently complicated due to the multiple resistance mechanisms it presents against different antibiotics of common use [10,11].

Fosfomycin is a phosphonic acid derivative that contains an epoxide and a propyl group; it is chemically analogous to phosphoenolpyruvate (PEP), which explains its mechanism of action, since it is a PEP competitor within the peptidoglycan synthetic pathway [12]. The enzyme MurA (UDP-N-acetylglucosamine enolpyruvyl transferase), which catalyzes the first step in peptidoglycan biosynthesis [13]—the transfer of enolpyruvate from PEP to uridine diphosphate N-acetylglucosamine (UDP-GlcNAc), is the only fosfomycin target known so far. Fosfomycin covalently binds to a cysteine residue in the active site of MurA, which renders MurA inactive. As a consequence of MurA inactivation, the peptidoglycan precursor monomers accumulate inside the cell; thus, peptidoglycan cannot be synthesized, and this leads to bacterial cell lysis and death [14].

Different molecular mechanisms leading to fosfomycin resistance have been described as being common in different organisms [15]. Firstly, allelic variants or mutations of MurA that do not contain a cysteine in their active site, as well as an increased synthesis of MurA, lead to fosfomycin resistance [13,16,17,18,19,20,21,22]. Secondly, the presence of an alternative route of peptidoglycan biosynthesis by means of recycling the peptidoglycan can avert the need for de novo synthesis through the MurA enzyme [23]. Thirdly, regarding mechanisms that involve a reduction in the intracellular concentration of the antibiotic, resistance can be achieved as the consequence of changes in the entry of fosfomycin to bacterial cells via mutations in any of the genes encoding the sugar phosphate transporters GlpT and UhpT, which are the gates for fosfomycin entry in various organisms [24,25]. In addition to an impaired transport inside the cell, the expression of efflux pumps could also reduce the intracellular concentration of fosfomycin. In fact, an efflux pump that confers resistance to fosfomycin, which is overexpressed in its presence, has been identified in *Acinetobacter baumannii* (AbaF) [26]. Fourthly, fosfomycin inactivation by fosfomycin-modifying enzymes such as FosA, FosB, and FosX [27,28,29,30] also renders resistance to this antibiotic.

It should be noted that a previous study has shown that these classical resistance mechanisms are not involved in the acquisition of increased fosfomycin resistance by *S. maltophilia*. This organism acquires increased levels of fosfomycin resistance as a consequence of mutations in genes encoding enzymes of the lower Embden-Meyerhof-Parnas (EMP) metabolic pathway [31]. In this lower glycolytic pathway, pyruvate is formed from glyceraldehyde-3-phosphate (GA-3P), at the same time as NADH and ATP are generated. The EMP pathway may also function in a gluconeogenic regime, forming hexose phosphate metabolites from a triose phosphate [32]. Remarkably, the inactivation of these enzymes does not cause major transcriptomic changes that could justify the observed resistance as the consequence of a collateral effect of the selected mutations on the expression of the aforementioned fosfomycin resistance mechanisms [31]. Rather, these mutations could produce alterations in the metabolic fluxes, affecting bacterial response in the presence of the antibiotic. In this regard, it is worth mentioning that intermediate metabolites can be toxic themselves. Indeed, previous studies carried out on *Escherichia coli* have shown that GA-3P has a bactericidal effect in Gram-negative bacteria, likely by blocking phosphoglycerate synthesis [33]. However, the mechanisms by which GA-3P inhibits cell growth have not been fully clarified. In a similar way, the accumulation of PEP causes the starvation of *Staphylococcus aureus* [34], and it has been suggested that it can inhibit mitochondrial respiration [35].

Given the interlinkage between fosfomycin resistance and *S. maltophilia*’s EMP metabolic pathway, in order to obtain further insight into the mechanisms of *S. maltophilia*’s response to fosfomycin, a transcriptomic study was carried out. Our goal was to determine whether the transcriptional changes caused by fosfomycin could be related to those due to the presence of two intermediate metabolites of this pathway, whose inactivation leads to fosfomycin resistance in *S. maltophilia*. The chosen metabolites were PEP—fosfomycin’s metabolic structural homolog—and GA-3P which, as mentioned above, is toxic to *E. coli*. With the aim of determining the contribution of stress-response networks in maintaining *S. maltophilia* homeostasis in the presence of fosfomycin, and whether the same networks are also relevant in facing the challenge of PEP and GA-3P, during this work we studied the transcriptional changes that this bacterium presents after subinhibitory treatment with the aforementioned compounds. The study of these transcriptomes provides new information about the crosstalk between metabolism and antibiotic resistance. This is the first study of the effects of fosfomycin and these two metabolites on the bacterial transcriptome in Gram-negative bacteria.

## 2. Results

### 2.1. S. maltophilia’s Susceptibility to Fosfomycin, PEP, and GA-3P

In order to compare the effects of the antibiotic fosfomycin and the EMP pathway metabolites PEP and GA-3P on the fitness of *S. maltophilia* D457, the growth rate of this microorganism was measured in the absence and presence of different concentrations of these compounds. Three concentrations below the minimum inhibitory concentration (MIC) of fosfomycin for *S. maltophilia* D457, which is 192 μg/mL [31], were selected to find the most suitable concentration for further study, so as to impact *S. maltophilia*’s fitness without fully impeding its growth (Figure 1A). MICs of PEP and GA-3P were also determined by double dilution to establish the proper subinhibitory concentrations to be used; MICs were 3800 μg/mL for PEP and 850 μg/mL for GA-3P. Taking into account our results, the concentrations chosen for further studies were 16 μg/mL for fosfomycin, 475 μg/mL for PEP, and 85 μg/mL for GA-3P (Figure 1B,C).

### 2.2. Effects of Fosfomycin, PEP, and GA-3P on S. maltophilia’s Transcriptome

As stated above, GA-3P is the initial/final compound of the route whose inactivation drives fosfomycin resistance; while PEP, apart from being part of the same metabolic route (Figure 1D), is analogous to fosfomycin, and is the natural substrate of the fosfomycin target MurA. Moreover, all three compounds inhibit *S.*
*maltophilia* growth (Figure 1). Comparing the global response of *S. maltophilia* to injury by each of these three compounds could provide clues as to the bacterial mechanisms of adaptation to the presence of fosfomycin, and to the interlinkage between fosfomycin activity and *S. maltophilia* metabolism. To enable the detection of primary and general responses of *S. maltophilia* to fosfomycin treatment, or to the addition of PEP and GA-3P as intermediate metabolites of the central metabolism, *S. maltophilia*’s transcriptome was analyzed in the presence and absence of said compounds.

The samples were processed, and the data obtained were analyzed as described in the Materials and Methods section. After 1 h of treatment with fosfomycin, statistically significant expression changes, which also presented fold changes of >1 log_2_ or <−1 log_2_ fold, were found in 228 genes, after PEP treatment in 440 genes, and after GA-3P treatment in 384 genes. The genes whose expression significantly changed after the treatments are detailed in Appendix A and Appendix A. Overall, the profiles of differentially expressed genes were similar between the different treatments, sharing 28.1% of the total changes (Figure 2). This percentage was even higher in the case of fosfomycin; 68% of the changes observed in the fosfomycin transcriptome also occurred in the other two transcriptomes, and just 16% of the changes were fosfomycin-specific.

The greatest proportion of differentially expressed genes belonged to the groups “stress response”, “chemotaxis and motility”, “transport”, and “metabolism”, indicating that the response obtained was mostly stress-associated. The number of genes with expression changes within each category in each transcriptome is detailed in Table 1.

Among the expression changes, it can be highlighted that the three compounds reduced the expression of genes related to transport across the membrane, such as the *smeZ* efflux pump gene and the fructose phosphotransferase system (PTS) (*ptsP*, *fruK*, *fruA* and *rpfN*); while upregulating those for amino acid biosynthesis (*aroG*, *hutH* or *hutI*), motility (*pilA*, *fliC*, etc.), chemotaxis (*cheA*, *cheW*, etc.), and stress response (*groES*, *sodA*, *dnaK*, *msrA* or *msrB*). Nevertheless, it is important to highlight the absence of transcriptional changes in genes related to cell wall biosynthesis, including *murA*, which encodes the cognate fosfomycin target, which is the biosynthetic pathway inhibited by fosfomycin. Moreover, transcriptional changes in genes related to the EMP pathway—which includes PEP and GA-3P as intermediate metabolites and has been proven to be involved in fosfomycin resistance—are not observed except from a phosphoglycerate mutase (Gpm)-encoding gene (SMD_RS04430), which is slightly downregulated after PEP and GA-3P exposure, with fold-change log_2_ ratios of −1.11 and −1.21, respectively, but not after fosfomycin treatment.

Apart from these common changes shared by the three treatments, expression changes observed after PEP and GA-3P treatments are almost twice as numerous as those found after fosfomycin treatment. This difference in the number of genes showing changes in their expression levels is mainly due to changes in the expression of genes encoding hypothetical proteins, ranging from 47 in the fosfomycin transcriptome to 111 and 86 in the PEP and GA-3P transcriptomes, respectively. Moreover, expression changes in genes encoding regulators increases from 18 after fosfomycin treatment to 31 in the PEP and GA-3P transcriptomes. Furthermore, PEP produced expression changes in 52 and GA-3P in 53 genes associated with replication, transcription, or translation, whereas in the fosfomycin transcriptome 38 genes of these categories experienced expression changes. The other important difference observed between the treatment with fosfomycin and the other two treatments is that some genes related to the electron transport chain—such as cytochrome C—are downregulated upon fosfomycin treatment, while others—such as *cioA*—are upregulated in the PEP and GA-3P transcriptomes. Finally, the xylose catabolism genes *xylA* and *xylB* are also upregulated after administration of PEP and GA-3P, ascending to 36 and 40 expression changes related to metabolism, respectively, whereas only 11 metabolism genes are found to suffer expression changes after fosfomycin treatment.

### 2.3. Stress Responses Are Strongly Affected by Fosfomycin, PEP, and GA-3P Treatments

In agreement with fosfomycin’s mechanism of action, the changes observed in the presence of the antibiotic are similar to those of the “cell-wall-stress stimulon”, which is a transcriptional response that has been described in *S. aureus* when subjected to inhibition of peptidoglycan biosynthesis by beta-lactams and fosfomycin [34,36,37]. It should be noted that PEP and GA-3P—common metabolic intermediates—also trigger the expression of the cell-wall-stress stimulon, suggesting that their presence in non-physiological conditions might interfere with peptidoglycan biosynthesis. Notably, all changes observed related to stress response in the transcriptome of *S. maltophilia* in the presence of fosfomycin were also found in either the PEP or the GA-3P transcriptome; a finding that further supports the interlinkage between fosfomycin resistance and *S. maltophilia* metabolism.

The induction of the cell-wall-stress stimulon includes the upregulation of different genes, among which are those that encode methionine sulfoxide reductase (MsrA) and the heat-shock protein GroES. In the case at hand, it should be noted that the three treatments induced the expression of the genes encoding MsrA, MsrB, and GroES, as well as the inhibition of general protein synthesis, as also described in the cell-wall-stress stimulon [37].

Many stress conditions induce the synthesis of heat-shock proteins, which mostly include molecular chaperones, such as those from the DnaK/DnaJ/GrpE and GroEL/GroES systems, and proteases, such as IbpA/IbpB and the ATP Lon-dependent proteases Clp ATPases and HslVU. Our data show that all treatments induce expression changes in all of the above except for the IbpAB protease. The reason behind the induction of these genes, and other stress-related genes, up to 25, 47, or 49 genes after fosfomycin, PEP, or GA-3P treatments (Table 2), respectively, may be the intracellular accumulation of damaged, misfolded, and aggregated cell-surface proteins, caused by the inhibition of cell wall biosynthesis. This accumulation would lead to the production of chaperones and proteases, such as *msrA* or *hslO*. In agreement with this, the induction of *hslO*, encoding the Hsp33 chaperone that is under heat-shock control at the transcriptional level, was observed. Moreover, the gene encoding the enzyme superoxide dismutase (SodA)—a defense enzyme against oxidative stress—was also upregulated.

In contrast to what has been described in previous studies, which maintain that chaperones are abundantly synthesized but protease levels are relatively low even in heat-shock induction conditions [38], our results show that the treatments produce a similar induction of chaperones and proteases, even obtaining the highest induction data in HslVU heat-shock proteases.

### 2.4. Motility, Chemotaxis, and Other Virulence-Related Genes Affected by the Three Different Treatments

Increased expression of motility and chemotaxis-related genes was detected; 28 expression changes in the case of fosfomycin treatment, and 39 and 34 in PEP and GA-3P treatments, respectively. Notably, expression changes in motility- and chemotaxis-related genes were not detected after fosfomycin treatment in *S. aureus*—the only species in which transcriptomic studies in the presence of fosfomycin have been published [34]. Due to this unique nature, to address the possibility that these expression changes might impact the production of elements relevant to virulence and infection, biofilm and swimming motility phenotypes were measured for bacteria growing on LB medium alone or containing either fosfomycin, PEP, or GA-3P. On the one hand, biofilm formation has been shown to be slightly enhanced in the presence of fosfomycin, PEP, or GA-3P; a physiological effect that is in concordance with the transcriptional results (Figure 3A). On the other hand, swimming motility did not show any statistically significant change after any of the three treatments (Figure 3B).

According to our transcriptomic data, PEP and GA-3P treatments also increased the synthesis of the type IV secretion system (T4SS) (Table 3), and although not reaching levels considered statistically significant, fosfomycin produced a similar effect. This secretion system includes a set of macromolecular transporters that can secrete proteins and DNA into the extracellular medium or into target cells [39]. To further confirm these expression changes, quantitative reverse transcription PCR (qRT-PCR) was performed. As shown in Figure 4, the expression of the T4SS is increased in the presence of PEP and GA-3P. This analysis also showed that even though expression changes were not considered statistically significant in the RNA-Seq analysis, fosfomycin was also able to induce the expression of the T4SS (Figure 4A).

*S. maltophilia*’s T4SS belongs to X-T4SSs—for Xanthomonadales-like T4SS group—a group that includes the T4SS homologs to that of *Xanthomonas citri*, which is involved in bacterial killing [39,40]. The T4SS channel includes a periplasmic core complex that forms a pore in the outer membrane that is made up of VirB7, VirB9, and VirB10, and is linked to the inner membrane via VirB10, which is known to play an important role in the regulation of substrate transfer to the extracellular space, along with an inner membrane complex composed of VirB3, VirB6, and VirB8, and three ATPases—VirB4, VirB11 and VirD4—and the extracellular pilus formed by VirB2 and VirB5 [39]. VirB10 forms the outer membrane pore, and traverses both the inner and outer membranes, being the scaffold protein of the T4SS. VirD4 plays an important role in recognizing substrates on the cytoplasmic face of the inner membrane and directing them for secretion through the T4SS channel. Proteins containing the conserved C-terminal XVIPCD domain (*Xanthomonas* VirD4-interacting proteins’ conserved domain) are antibacterial effectors secreted via the T4SS channel into the target cell; these proteins carry N-terminal domains with enzymatic activities predicted to target structures in the cell envelope [41]. A bioinformatic search on the *S. maltophilia* D457 genome identified 16 putative T4SS substrates (Appendix A). Notably, one of these potential effector proteins shows an expression change in the three transcriptional studies. This effector, SMD_RS14045, is predicted to be a hydrolase—formimidoylglutamate deiminase—involved in L-histidine metabolism; it may be recognized by VirD4 and secreted by this T4SS, playing a putative role in secretion functions, in addition to its known metabolic activity (Table 4).

As mentioned, T4SS is a virulence apparatus that delivers effectors into eukaryotic or bacterial target cells. This system enhances the growth of *S. maltophilia* when it is co-cultured with different bacteria. Previous works have shown that *S. maltophilia* T4SS activity may produce the death of *E. coli*, *Pseudomonas aeruginosa*, and *Pseudomonas mendocina* in a contact- and concentration-dependent manner [39,40]. To test whether these expression changes correlate with an enhancement of *S. maltophilia* growth when co-cultured with a competitor, bacterial competition assays—in which *S. maltophilia* was pre-cultured with either fosfomycin, PEP, or GA-3P, and subsequently washed—were performed. After 24 h of competition with *E. coli* and *P. aeruginosa* PA14, preincubation with either fosfomycin, PEP, or GA-3P allows an increase in the amount of *S. maltophilia* present in the mixed cultures, in comparison with the control cultures performed in the absence of these compounds (Figure 5). However, it is also worth mentioning that these treatments did not produce any change in the *S. maltophilia* D457/*P. aeruginosa* PAO1 relationship with respect to the control conditions, indicating that this effect might depend not only on the competitor species, but also on the competing strain. These results suggest that the tested compounds are able to enhance the VirB/D4 T4SS, resulting in a better fitness of *S. maltophilia* in the presence of other bacteria by impeding the growth and/or survival of some competing strains. It is important to highlight that these competitors share niches—such as cystic fibrotic lungs—with *S. maltophilia*, suggesting that the presence of these compounds in a niche of infection could afford an adaptive advantage for *S. maltophilia* against some competitors.

### 2.5. Metabolic Pathways Affected by Fosfomycin, PEP, and GA-3P Treatments

Apart from the overexpression of proteins related to the stress response, the activation of the cell-wall-stress stimulon includes the overexpression of the enzyme IIA of the PTS. The PTS mediates the uptake and phosphorylation of the D-configuration of carbohydrates by transferring a phosphoryl from PEP to a histidine residue [42], thus controlling metabolism in response to the availability of carbohydrates. The sugars that are commonly transported across the cell membrane by the PTS are glucose, mannose, fructose, the hexitols, N-acetylglucosamine, N-acetylmannosamine, glucosamine, mannosamine, and β glucosides [43]. Unlike the cell-wall-stress stimulon response, the PTS that transports fructose—specifically fructose-specific IIBC components (FruA)—is strongly repressed by fosfomycin, PEP, and GA-3P treatments in *S. maltophilia* (Table 4 and Figure 4B). Expression of *SMD_RS11695* is also inhibited after the three treatments, and even though *SMD_RS11695* is proposed to encode the nitrogen-metabolic PTS PtsP subunit—involved in regulating nitrogen metabolism [44]—its structure resembles that of the FruB protein encoded by *Pseudomonas putida*. The PTS fructose-only transport system from *P. putida* involves two proteins: the EIIBC membrane embedded and encoded by *fruA*, and EI/HPr/EIIA^Fru^, which is cytoplasmic and encoded by *fruB* [45]. *P. putida* FruB contains the same domains as SMD_RS11695, which is composed of EIIA^Fru^ from residues 11 to 148, Hpr from 165 to 243, and EI from 278 to 809 [46], suggesting that *S. maltophilia*’s fructose PTS encodes its own EI, and that SMD_RS11695 might be renamed *fruB*. Moreover, apart from a decrease in the intracellular concentration of fructose 1-phosphate due to PTS downregulation, *fruK*—needed to form the EMP intermediate fructose 1,6-phosphate from fructose 1-phosphate—is also highly repressed after the three treatments.

Likewise, the three transcriptomes analyzed here show that the presence of either fosfomycin, PEP, or GA-3P produces a common inhibition of genes related to lactate metabolism; intracellular transport of lactate is inhibited, as is its oxidation to pyruvate (Table 4 and Figure 4C). Nevertheless, with respect to amino acid metabolism, increased expression of genes related to L-histidine degradation is observed, leading to an increase in L-glutamate concentration (Table 4), as well as an increased L-arginine biosynthesis—especially after PEP and GA-3P treatments.

### 2.6. Effects of Fosfomycin, PEP, and GA-3P on the Central Metabolism: Enzymatic Activity of the Main Dehydrogenases and Lower Glycolysis Enzymes

The expression of genes related to central metabolism—except for one of the two phosphoglycerate mutase genes encoded in *S. maltophilia*’s genome—did not change according to our transcriptomic assay (Table 5). However, transcription is not the only way to regulate metabolic activity. Hence, to elucidate whether the antibiotic fosfomycin, its metabolic mimic PEP, and the glycolytic intermediate metabolite GA-3P may produce relevant metabolic shifts in *S. maltophilia*’s physiology, the enzymatic activities of the main dehydrogenases of the central metabolism were measured after treatment with fosfomycin, PEP, or GA-3P. These activities are good indicators of the general physiological state of the cell, including its redox balance. In particular, the activities of the glucose-6-phosphate dehydrogenase (Zwf)—which connects the glucose-6-phoshpate with the Entner–Doudoroff (ED) and pentose phosphate (PP) pathways—and the isocitrate dehydrogenases Icd NAD+ and Icd NADP+ from the tricarboxylic acid (TCA) cycle were determined. The activity of the enzyme Zwf increased by 2.5–3.3-fold after the three treatments, as compared with the wild-type D457 strain without treatment, whereas the activity of both the Icd NAD+ and NADP+ enzymes did not change in any of the groups (Figure 6).

In addition, the activity of the lower glycolysis enzymes was measured after the three treatments. This amphibolic metabolic pathway (the EMP pathway) includes PEP and GA-3P as intermediate metabolites, and its inactivation has been proven to be the cause of fosfomycin resistance in *S. maltophilia* [31]. As shown in Figure 7, glyceraldehyde-3-phosphate dehydrogenase (Gap), phosphoglycerate kinase (Pgk), Gpm, enolase (Eno), and pyruvate kinase (Pyk) activities decreased after the three treatments, by 2–8.4-fold. A diagram of the central metabolism of *S. maltophilia* is represented in Figure 8. As shown, the treatment of *S. maltophilia* with any of the three compounds reduced the activity of enzymes whose inactivation causes fosfomycin resistance in this organism [31]. This finding further supports the notion that exposure to sub-MIC concentrations of either fosfomycin, PEP, or GA-3P triggers a similar stress response that enables *S. maltophilia* to maintain cellular homeostasis in the presence of these compounds.

### 2.7. General Transport Is Downregulated by Fosfomycin, PEP, and GA-3P

Changes in the expression of multiple transporters were observed, in addition to the aforementioned repression of the fructose PTS. Although the general trend is a repression of both influx and efflux transport, expression of different TonB transporters is induced, as well as magnesium influx—especially in the PEP and GA-3P transcriptomes.

Apart from using sugar phosphate transporters to cross the inner membrane, fosfomycin can enter *P. aeruginosa* PAO1 through the outer membrane via OprP, and more rapidly through OprO [47], which are phosphate-uptake-facilitating porins. Both porins allow the high-affinity uptake of phosphate anions—important for bacterial growth—and they are induced under phosphate-starvation conditions. These porins, also described in other microorganisms, were found to be orthologous of SMD_RS18370 and SMD_RS18400 in *S. maltophilia* D457. All treatments carried out in our study produced the inhibition of the expression of *SMD_RS18400* (Table 6 and Figure 4D), which presents a protein identity of 28% with OprO and 25% with OprP from *P. aeruginosa* PAO1. The last fact to highlight in relation to outer membrane transport, and referring to what was previously explained, is the inhibition of the expression of the RpfN porin, which is part of the PTS fructose transport operon, and shows homology with the OprB porin, which is able to transport carbohydrates to the periplasmic space [48] (Table 3).

Multidrug resistance (MDR) efflux pumps are major contributors to drug resistance in *S. maltophilia*; among them, the best characterized group is the resistance–nodulation–division (RND) family. One of these complexes is the SmeYZ efflux pump, which is constitutively expressed, and contributes to intrinsic resistance to aminoglycosides, tetracycline, leucomycin, and trimethoprim/sulfamethoxazole (SXT). Therefore, the observed changes in the expression levels of *smeYZ* (*SMD_RS10440* and *SMD_RS10445*) after the three treatments should be highlighted. A decrease in the expression of this efflux pump was observed in the three transcriptomes, as shown in Table 6 and verified by qRT-PCR (Figure 4D), implying that fosfomycin, PEP, and GA-3P are the first described inhibitors of this efflux pump. To address the functional consequences of these findings, the susceptibility of *S. maltophilia* to antibiotics transported outside the cell by SmeYZ was measured in the presence of fosfomycin, PEP, and GA-3P. The results show that the tested compounds are able to increase *S. maltophilia*’s susceptibility to the antibiotics pumped out by SmeYZ (Table 7). These findings support the possibility that fosfomycin, PEP, and GA-3P could be used as adjuvants of antibiotics with relevance for treating *S. maltophilia* infections, such as aminoglycosides and SXT.

## 3. Discussion

The exposure of bacteria to antibiotics produces deep changes in bacterial physiology that are associated with alterations in bacterial transcriptomes. While some of these alterations are indirect consequences of the presence of the antimicrobials, other transcriptional changes are primarily associated with the expression of stress-related genes that serve for maintaining bacterial homeostasis in the presence of the drug. For instance, transcriptional studies in *S. aureus* have served to define a set of genes whose expression changes after inhibition of peptidoglycan synthesis by treatment with antibiotics that inhibit this synthesis, but not with antibiotics that affect other cellular targets [36]. In this way, a characteristic cell response that enables the microorganism’s defense against the inhibition of the synthesis of peptidoglycan was defined; this response is called the cell-wall-stress stimulon. The induction of the cell-wall-stress stimulon was defined after the treatment of *S. aureus* with beta-lactams and this same response was subsequently described after treatment of *S. aureus* with fosfomycin. The cell-wall-stress stimulon includes, among others, the following proteins, whose expression is increased in the presence of the stressors [34,37]: (1) the MsrA enzyme; (2) the TRAP transcription signal protein, involved in the regulation of RNA III production; (3) the transcriptional elongation factor GreA; (4) the heat-shock protein GroES; and (5) the IIA enzyme of the PTS system. In addition to the overexpression of these enzymes, the inhibition of general protein synthesis was observed [37]. It is important to note that the *S. aureus* transcriptional study is the only study published to date that elucidates the effects of fosfomycin on gene expression.

The enzyme MsrA is classified within the category of chaperones and posttranslational-modification-related determinants, with the inhibition of peptidoglycan biosynthesis being a requirement for its induction [49]. Our results show that not only fosfomycin, but also PEP and GA-3P—not known to interfere with peptidoglycan biosynthesis—induce the expression of *msrA* in *S. maltophilia*. These methionine sulfoxide reductases are defense proteins against oxidative stress; they reduce methionine sulfoxide residues to methionine and, thus, restore protein function [49]. In agreement with their connection with an oxidative stress response, *sodA* is also overexpressed. Moreover, both MsrA and SodA need thioredoxin reductases for the regeneration of their active forms. These reductases use NADPH as a cofactor, which may correlate with the described increase in Zwf activity during treatments. Zwf is a key enzyme in the central metabolism that transforms glucose-6-phosphate into 6-phosphogluconate, and generates NADPH via its catabolic action. Due to the production of NADPH, this enzyme is involved in the defense against cellular stresses, such as oxidative stress [50]. The increased activity of Zwf, along with MsrA and SodA overexpression, could be related to possible detoxification of the compounds that would occur analogously to ROS detoxification.

Moreover, chaperones and proteases are essential for de novo folding and protein quality control, preventing protein aggregation and folding back or degrading poorly folded proteins [51]. Moreover, two important functions for cell survival have traditionally been attributed to molecular chaperones: (a) prevention of unfolded protein aggregation, and (b) assistance in the correct refolding of the denatured polypeptides attached to the chaperones [52,53,54]. Due to the fact that fosfomycin and the other two metabolites do not directly generate aberrant or misfolded proteins, it could be possible that the expression of heat-shock chaperones may contribute to resistance against these compounds after cell wall inhibition, stabilizing the enzymes that synthesize components of the cell wall, as seen in *Streptococcus pneumoniae* in the case of beta-lactam treatments [39].

*S. aureus* and other Gram-positive organisms harbor two UDP-N-acetylglucosamine enolpyruvyl transferase genes (*murA* and *murZ*). MurA serves as the major UDP-GlcNAc enolpyruvyl transferase under normal growth conditions, in which its expression is five times greater than that of MurZ, but its expression is not inducible. In contrast to MurA, expression of the MurZ enzyme is inducible when the rate of peptidoglycan production is insufficient [55]. Previous analysis showed that *murZ* was overexpressed in *S. aureus* in the presence of fosfomycin, while *murA* was not significantly overexpressed under the same conditions [34]. In contrast, no significant differences in expression were observed in any of the genes involved in the synthesis of peptidoglycan in our study, indicating that the adaptation of *S. maltophilia* to the presence of fosfomycin does not require changes in the activity of classical determinants involved in the activity of and resistance to this antibiotic.

Apart from the absence of changes in genes related to the canonical route of action of fosfomycin, loss-of-function mutations of the genes encoding lower glycolysis enzymes have been previously found to be the cause of fosfomycin resistance in *S. maltophilia* [31]. However, expression changes in this fosfomycin target route were only observed after both PEP and GA-3P treatments, which are able to inhibit the expression of a Gpm lower glycolysis enzyme (SMD_RS04430). Despite the absence of transcriptional changes apart from those previously mentioned, all tested compounds—fosfomycin included—were able to inhibit the enzymatic activity of the lower glycolysis enzymes. The inhibition of these enzymatic activities could be a feedback mechanism of resistance to deal with the toxicity caused by these compounds. Together with previous information derived from the analysis of stable fosfomycin-resistant mutants [31], these results reinforce the linkage between primary metabolism and fosfomycin resistance in *S. maltophilia*. Furthermore, these results may suggest that, in addition to inhibiting peptidoglycan biosynthesis, fosfomycin might have a secondary effect on the central carbon metabolism—a feature that remains to be explored.

The last step of the induction of the cell-wall-stress stimulon is the overexpression of the IIA enzyme of the PTS system. Previous studies propose that IIA induction increases glucose transport within the cell and, thus, provides the energy necessary to increase peptidoglycan biosynthesis [26]. However, unlike what has been described so far, the PTS system that transports fructose was strongly repressed by our treatments; it could thus be a defense system against the transport of the toxic compounds within the cell. Each PTS transporter is made up of sugar-specific proteins—two cytoplasmic domains (EIIA and EIIB) and a membrane-integrated domain (EIIC)—and general proteins—enzyme I (EI) and HPr, which transfer the phosphoryl group from PEP to the sugar-specific proteins. The fructose PTS does not follow these general PTS conformations; it has been described as being formed by two different proteins in most bacteria: FruA and -B, also known as the Enzyme II^Fru^ complex. This complex contains a unique combination of PTS domains, with three domains in the FruA (IIB’-IIB-IIC) protein and three domains in the FruB protein (IIA-M-H). In the FruB protein, domain M is a central domain of an unknown function, and H—also known as pseudo-HPr—substitutes for HPr in the phosphoryl transfer reaction [56]. *P. putida*’s fructose PTS is an exception to this conformation because, as has been previously mentioned, it presents a FruB protein containing an EI domain in addition to the IIA and HPr domains [45]. Previous studies have proven that mutations in general PTS proteins can lead to disparate phenotypes concerning susceptibility to fosfomycin; *ptsI* mutants are fosfomycin-resistant, but *ptsH* mutants are fosfomycin-susceptible [43]. It has been suggested that the resistance phenotype given by mutations in this system has indirect effects in other systems—especially in systems capable of transporting fosfomycin [43]—which could be on the basis of the transcriptional response observed in our analysis. Our results show that fosfomycin, PEP, and GA-3P inhibit the expression of the EIIBC sugar-specific complex of the fructose PTS (FruA) as well as a possible fructose-specific EI/HPr/EIIA^Fru^ complex of the fructose PTS (FruB), by which fructose and other sugars, such as glucosamine, cross the inner membrane. Fructose is internalized as fructose 1-phosphate by FruB and FruA, and then it is converted to fructose 1,6-bisphosphate by FruK, the expression of which is also downregulated by the analyzed stressors, leading to a lower concentration of fructose 6-phosphate. The FruA transporter also uses part of the UDP-GlcNAc biosynthetic pathway. GlcNAc is an essential precursor of cell wall peptidoglycan, and is a substrate of the enzyme MurA, which catalyzes the transfer of enolpyruvate from PEP to UDP-GlcNAc as a first step in peptidoglycan biosynthesis, and is the only known fosfomycin target [13]. The first committed step in UDP-GlcNAc biosynthesis forms glucosamine 6-phosphate from fructose 6-phosphate. However, the product of this first reaction—glucosamine 6-phosphate—can also be transported into the cell and used as a carbon source thanks to the PTS fructose transporter FruA [57]. Therefore, the decreased expression of FruA would lead not only to a decreased concentration of UDP-GlcNAc, but also to poor peptidoglycan biosynthesis and an increase in the intracellular concentration of PEP. The accumulation of PEP by the suppression of the PTS system—as with the inhibition of MurA—acts as a carbon starvation signal [58]. In response to the lack of carbon, repression of energy metabolism, amino acid degradation, and the TCA cycle would be expected [34]; however, this response was not conclusively observed in our analysis.

Apart from this cell-wall-stress stimulon related response, overexpression of virulence-related genes and chemotaxis or secretion system genes, along with increased expression of genes involved in the synthesis of L-arginine and L-glutamate, has been observed in the presence of the studied stressors. Overexpression of genes related to D-glutamate production—such as *murI* and *gltD*—as a cellular response after inhibition of peptidoglycan synthesis has been previously described in the case of *S. aureus* [36]. Moreover, L-glutamate takes part in the GlcNAc biosynthetic pathway, and it is needed to synthetize glucosamine 6-phosphate [57]. The increase in L-glutamate could be related to the lack of transport of glucosamine 6-phosphate by FruA, thus increasing the concentration of this essential product for the synthesis of UDP-GlcNAc. In addition, a generalized repression of the expression of tRNAs—directly connected with the synthesis of amino acids—was also observed in our study. Regarding energy metabolism, a general response was not detected, as we observed a greater amount of expression changes in response to PEP and GA-3P than to fosfomycin, but without a clear tendency to repression.

In addition to the fructose PTS transporter, other transporters are downregulated in the presence of fosfomycin, PEP, and GA-3P, such as SmeYZ. SmeYZ is an efflux pump that contributes not only to intrinsic resistance in *S. maltophilia*, but also to its acquired and phenotypic resistance to different antibiotics [59,60,61]. While inducers of the expression of this efflux pump have been described [61], inhibitors of the expression of efflux pumps are rarely found, despite the fact that they could be useful adjuvants for sensitizing bacteria expressing these determinants of antibiotic resistance. Our results suggest that fosfomycin, PEP, or GA-3P could be used in combination with the substrates of SmeYZ, reducing its expression and, hence, increasing susceptibility to these antibiotics. Further studies are needed in order to ascertain the best inhibitor, which could be used in clinics for the treatment of infections caused by *S. maltophilia*. As a matter of fact, SXT is the treatment of choice for treating *S. maltophilia* infections [62], so combination therapies of SXT and fosfomycin could exhibit a greater activity than SXT alone.

The fosfomycin transporters UhpT and GlpT—glucose-6-phosphate and glycerol-3-phosphate transporters, respectively—are not encoded in the *S. maltophilia* genome, and previous studies have suggested that *S. maltophilia* lacks any canonical fosfomycin transporter [31]. However, the observed inhibition of OprO and RpfN by sub-MIC fosfomycin concentrations that, as discussed above, trigger a stress-protective response, could indicate that fosfomycin reaches the periplasmic space via these porins. This fact, as well as the observed inhibition of FruAB expression, could impart new insights for the identification of *S. maltophilia* fosfomycin transporters.

All in all, the present study enabled us to determine the response of *S. maltophilia* to fosfomycin, as well as its analog PEP and the first/last intermediate metabolite of the route whose inactivation causes fosfomycin resistance (GA-3P). The three responses were similar, and shared expression changes with those of the cell-wall-stress stimulon previously observed after administration of antibiotics that interfer with the cell wall in *S. aureus*. However, in contrast to previous findings, cell-wall-stress genes—including *mur* genes—did not show any expression changes. Apart from the overexpression of stress-related genes, virulence-related genes and amino acid metabolism genes were also overexpressed. In addition to these changes, translation and several transport pathways were downregulated, and a response explained by a starvation response was not observed.

Altogether, our results indicate that in addition to being a PEP competitor for MurA binding, fosfomycin mimics the effects of two metabolites—PEP and GA-3P—on *S. maltophilia*’s transcriptome. In addition, we found that the three compounds improved the fitness of *S. maltophilia* when growing in the presence of competitors, reducing its resistance to antibiotics in common use. These results support the interlinkage between antibiotic resistance, virulence, and bacterial metabolism [63], providing useful information for the implementation of metabolic interventions to tackle antibiotic resistance [64].

## 4. Materials and Methods

### 4.1. Bacterial Strain and Culture Conditions

The *S. maltophilia* D457 strain, originally obtained from a bronchial aspirate [65,66], was used in this study. Unless otherwise stated, bacteria were grown in LB (lysogeny broth) Lennox medium at 37 °C, with constant agitation at 250 rpm. Solid media were prepared using an agar concentration of 15 g/L. Different concentrations of fosfomycin, PEP, and DL-GA-3P were used in different experiments, as stated in the different sections.

### 4.2. Bacterial Growth Measurement

*S. maltophilia* D457’s MIC values for PEP and DL-GA-3P were obtained on Mueller–Hinton (MH) medium by double dilution after 48 h at 37 °C. *S. maltophilia* D457 cells were grown in LB medium at 37 °C, with different compounds used at serial concentrations. The compounds used were fosfomycin (64, 32, and 16 μg/mL), PEP (1900, 950, and 475 μg/mL), and DL-GA-3P (340, 170, and 85 μg/mL). The stock solutions of the different compounds were diluted in LB medium to obtain the required concentrations. Growth was measured with a Spark 10M plate reader (Tecan, Männedorf, Switzerland) at OD_600_ in flat-bottomed transparent 96-well plates (Nunc MicroWell; Thermo Fisher; Waltham, MA, USA). Then, 10 μL of cell culture was inoculated in 140 μL of medium in each well, to a final OD_600_ of 0.01. The plates were incubated at 37 °C with 10s of shaking every 10 min. In all cases, a non-inoculated well containing the corresponding medium was included as a test of medium sterility.

### 4.3. Protein Quantification

Protein concentration was determined following the Pierce BCA Protein Assay Kit (Thermo Scientific) protocol in 96-well plates (Nunc MicroWell; Thermo Fisher).

### 4.4. In Vitro Activity Assays of the Enzymes of the Lower Glycolytic Pathway and Dehydrogenases

Cells were grown until the exponential growth phase (OD_600_ = 0.6), when 16 μg/mL fosfomycin, 475 μg/mL PEP, or 85 μg/mL GA-3P was added. A culture without treatment was used as a control. After 1 h of treatment at 37 °C, cells were harvested by centrifugation at 5100× *g* and 4 °C, and then washed twice in 0.9% NaCl and 10 mM MgSO_4_. Once washed, cells were disrupted by sonication at 4 °C, and the cell extracts were obtained by centrifugation at 23,100× *g* for 30 min at 4 °C.

NAD(P)+ reduction and NAD(P)H oxidation were monitored spectrophotometrically at 340 nm and 25 °C with intermittent shaking in microtiter plates, using a Spark 10M plate reader (Tecan). Each reaction was performed using three biological replicates, and the specific activities were obtained by dividing the measured slope of NAD(P)H formation or consumption by the total protein concentration. Enzymatic activities of dehydrogenases (glucose-6-phosphate, isocitrate NAD+, isocitrate NADP+, and glyceraldehyde-3P dehydrogenases) were measured as described previously [31].

Enzymatic activities of phosphoglycerate kinase, phosphoglycerate mutase, and enolase were assayed following the protocol described by Gil-Gil et al. [31], in a two-step reaction. Pyruvate kinase was measured as stated by Pawluk et al. [67], with some modifications. Pyruvate kinase activity was determined in a first step by adding 10 μL of the cell extract to 90 μL of K/MES buffer at pH 6.5 (including 30 mM KCl and 3mM MgCl_2_), 0.5 mM ADP, and 0.5 mM phosphoenolpyruvate. After 15 min of incubation at room temperature, the mixtures were heated for 1 min at 95 °C to stop the reaction. In a second step, the formation of lactate was measured by adding 0.15 mM NADH and 10 units of lactate dehydrogenase/mL. NADH oxidation was monitored spectrophotometrically.

### 4.5. RNA Extraction and RNA-Seq

*S. maltophilia* was grown overnight in LB broth at 37 °C and 250 rpm. This culture was used to inoculate new flasks, to reach an optical density at 600 nm (OD_600_) of 0.01, and the cultures were grown at 37 °C until an OD_600_ of 0.6 was reached. When bacteria grew in the exponential phase (OD_600_ ≈ 0.6), the transcriptional response of *S. maltophilia* to treatment with 16 μg/mL fosfomycin, 475 μg/mL PEP, or 85 μg/mL GA-3P was evaluated by adding the compounds to the bacterial cultures for 1 h at 37 °C. A culture without treatment was used as the expression control. Afterwards, RNA was isolated following the protocol described by Gil-Gil et al. [31]. DNA contamination was checked by PCR with primers 27 and 48 (Appendix A). Only RNAs containing no DNA contamination were used for further studies. RNA sequencing (RNA-Seq) was conducted by Macrogen, Inc. Libraries were constructed with the TruSeq Stranded Total RNA Library, and rRNA was depleted with the NEBNext rRNA Depletion Kit. Sequencing was conducted with NovaSeq6000 sequencing technology, using a 150 bp paired-end format and 20 million total reads/sample, with three independent biological replicates of each sample. The quality of paired-end short reads (in FASTQ format) was evaluated with FASTQC [68]. Traces of Illumina adapters were detected at the 3’ end, so reads were cropped in the alignment process. Reads were aligned against the *S. maltophilia* genome (NC_017671) using RNA-STAR [69] (--alignIntronMax 1 --clip3pNbases 50). Optical and PCR duplicates were identified with the “MarkDuplicates” function of the GATK suite [70]. *S. maltophilia* genes were quantified using the featureCounts function of the Bioconductor [71] package Rsubread [72] (strandSpecific = 2,isPairedEnd = TRUE, requireBothEndsMapped = TRUE, primaryOnly = TRUE, ignoreDup = TRUE). Alignments were visualized with the IGV browser [73]. Differential expression between sample groups and their statistical significance were calculated with the Bioconductor package DESeq2 [74], using the function lfcShrink (type = “normal”, cooksCutoff = FALSE, independentFiltering = TRUE, lfcThreshold = 0, alpha = 0.05). Gene annotations were added to final expression files and converted to spreadsheet files (xlsx) with R. Only fold changes of < −1 or > 1 were considered to be relevant values. A Boolean analysis of RPKM fold-change-relevant values for each gene, for each treatment, relative to the strain without treatment, was performed using the Venny tool [75].

### 4.6. Quantitative Reverse Transcription PCR (qRT-PCR)

Using a high-capacity cDNA reverse transcription kit (Applied Biosystems, Waltham, MA, USA), cDNA was obtained from 10 g of RNA. qRT-PCR was carried out with a Power SYBR Green PCR Master Mix (Applied Biosystems) in an ABI Prism 7500 real-time system (Applied Biosystems). A total of 50 ng of cDNA was used in each reaction, except for the wells with no template, which were used as negative controls. A first denaturation step at 95 °C for 10 min was followed by 40 cycles at 95 °C for 15 s and 60 °C for 1 min, for amplification and quantification. Primers that amplify specific fragments of the desired genes were designed with Primer3 Input software, and were used at 400 nM (Appendix A). The primers gyrA_F and gyrA_R were used to amplify the housekeeping gene *gyrA*. Differences in the relative amounts of mRNA were determined according to the 2^−ΔΔCT^ method [76]. In all cases, the values of relative mRNA expression were determined as the average of three independent biological replicates, each containing two technical replicates.

### 4.7. Biofilm Formation Assay

An overnight culture of *S. maltophilia* D457 was diluted 1:100 in LB medium or LB medium with 16 μg/mL fosfomycin, 475 μg/mL PEP, or 85 μg/mL GA-3P. A total of 100 μL of bacterial suspension was inoculated per well in a 96-well plate (Costar Serocluster^TM^, Corning Incorporated). After 48 h of incubation at 37 °C without agitation, biofilms were stained by adding 25 µL of 0.1% crystal violet for 5 min. The stained biofilms were rinsed three times using 100 µL of Milli-Q water, and then 150 µL of 0.25% Triton X-100 was added in order to dissociate the biofilms; 100 µL was then transferred to a clean 96-well plate (Nunclon^TM^ Delta Surface), and biofilm formation was assessed via the quantification of crystal violet staining, by measuring absorbance at 570 nm. The assay was performed in octuplicate.

### 4.8. Swimming Assay

The swimming motility of *S. maltophilia* D457 was determined on LB agar (0.3%) plates and LB agar plates (0.3%) with 16 μg/mL fosfomycin, 475 μg/mL PEP, or 85 μg/mL GA-3P. An overnight culture from D457 was diluted to a final OD_600_ of 2, and 5 µL was spotted on the surface of the swimming plates. After 48 h of incubation at 30 °C with humidity, the growth zone was measured in centimeters. The assay was performed in triplicate.

### 4.9. Bacterial Competition Assays

Bacterial competition was assessed by analyzing target cell growth following previously described methods [40], with some modifications. To analyze *E*. *coli* K12, *P. aeruginosa* PA01, and *P. aeruginosa* PA14 growth during co-incubation with *S*. *maltophilia*, each strain was sub-cultured (OD_600_ = 0.01) and grown to the exponential phase (OD_600_ = 0.6) at 37 °C and 250 rpm. When the exponential growth phase was achieved, 16 μg/mL fosfomycin, 475 μg/mL PEP, or 85 μg/mL GA-3P was added to *S. maltophilia* D457 bacterial cultures for 1 h at 37 °C, and subsequently washed twice by centrifugation at 5100× *g* in LB medium. Using fresh LB medium, each bacterial suspension was then diluted to an OD_600_ of 0.3, providing approximately equivalent CFUs/mL. Aliquots of the cell suspensions were then combined so as to obtain ratios of *S. maltophilia* to the heterologous bacterium of around 200:1. Then, 100 µL of serially diluted mixtures was spotted on LB agar plates and 25 μg/mL imipenem–LB agar plates, and incubated at 37 °C for 24 h to obtain a control of the initial proportion of each bacterium. The colonies formed on the antibiotic-containing plates represented the abundance of *S. maltophilia* in the cell mixture, whereas the colonies on the standard plates reflected the total cell population. Moreover, 50 µL of each mixture was spotted on LB agar plates and incubated at 37 °C for 24 h in order to measure the effects on bacterial growth. After incubation, the area of bacterial growth of these plates was removed and resuspended in 1 mL of 0.85% NaCl. The resultant cell suspensions were serially diluted and plated onto both LB agar and 25 μg/mL imipenem–LB agar plates. As previously indicated, the colonies formed on the imipenem plates represented the abundance of *S. maltophilia* in the cell mixture. The abundance of heterologous bacteria surviving in the co-culture was determined by calculating the difference in CFUs between the two plates.

### 4.10. Antimicrobial Susceptibility Assays

The amikacin, tobramycin, gentamicin, tetracycline, and SXT MICs were determined for the D457 strain on LB agar, LB agar with 16 μg/mL fosfomycin, LB agar with 475 μg/mL PEP, and LB agar with 85 μg/mL GA-3P, using MIC test strips (MIC Test Strips; Liofilchem Diagnostics, Roseto degli Abruzzi, Italy). Plates were incubated at 37 °C, and the results were analyzed after 20 h. The experiments were performed in triplicate.

### 4.11. BLASTp Search

To identify putative effectors secreted by the T4SS of *S. maltophilia* D457, we used the XVIPCDs of known and putative *X. citri* T4SS substrates (residues in parentheses): XAC4264(140–279), XAC3634(189–306), XAC3266(735–861), XAC2885(271–395), XAC2609(315–431), XAC1918 (477–606), XAC1165(1–112), XAC0574(317–440), XAC0466(488–584), XAC0323(16–136), XAC0151(120–254), and XAC0096(506–646). A BLAST search of these substrates against the genome of *S. maltophilia* D457 (https://www.genome.jp/tools/blast/) was carried out. To identify porins in the genome of *S. maltophilia* D457, a BLAST search was carried out with the protein sequence of the *P. aeruginosa* PAO1 porins.

## Figures and Tables

**Figure 1 ijms-23-00159-f001:**
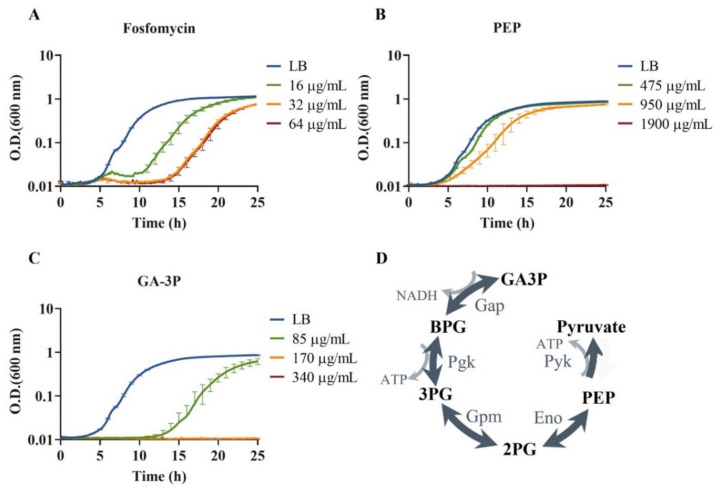
Effects of (**A**) fosfomycin, (**B**) PEP, and (**C**) GA-3P on the growth of *S. maltophilia* D457 in LB medium. Growth curves were measured in the presence of different compounds’ concentrations, and growth in LB medium was used as a control. Error bars indicate standard deviations for the results from three independent replicates. Panel (**D**) shows the metabolic pathway where previously identified mutations leading to fosfomycin resistance in *S. maltophilia* are located [31]. Substrate abbreviations—GA3P: glyceraldehyde-3-phosphate; BPG: 1,3-bisphosphoglycerate; 3PG: 3-phosphoglycerate; 2PG: 2-phosphoglycerate; PEP: phosphoenolpyruvate. Enzymes: Gap: GA3P dehydrogenase; Pgk: phosphoglycerate kinase; Gpm: phosphoglycerate mutase; Eno: enolase; Pyk: pyruvate kinase.

**Figure 2 ijms-23-00159-f002:**
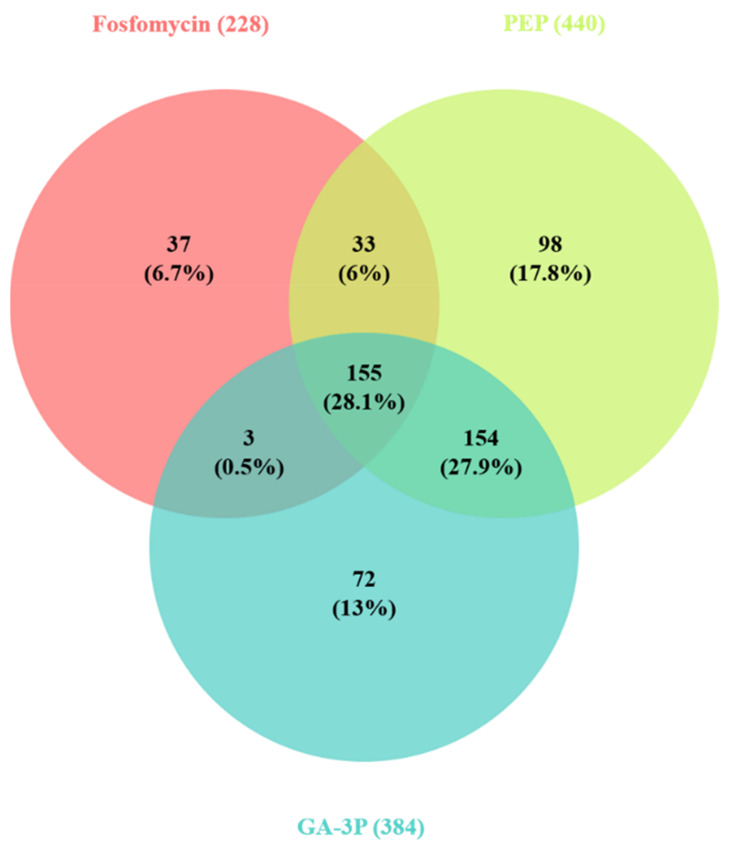
Common and differential transcriptomic changes after *S. maltophilia* treatments. Venn diagram showing the number of genes whose expression changed (>1 log_2_ fold or <−1 log_2_ fold) after the treatments. As shown, most transcriptomic changes were common between the treatments.

**Figure 3 ijms-23-00159-f003:**
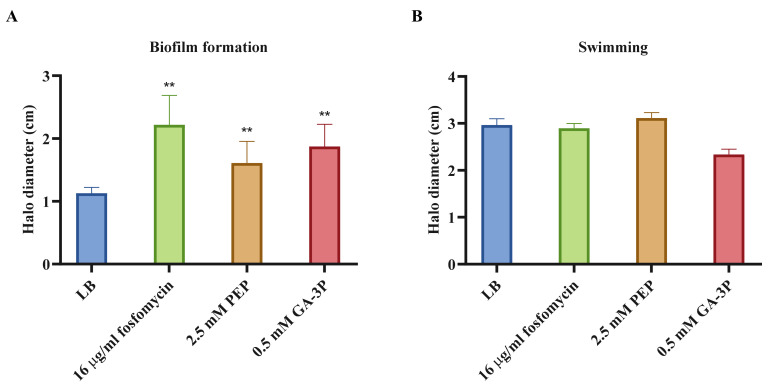
Quantification of chemotaxis- and motility-related phenotypes relevant to the virulence of *S. maltophilia* after fosfomycin, PEP, and GA-3P treatments. The graphs show the (**A**) biofilm formation and (**B**) swimming motility of *S. maltophilia*. Error bars indicate standard deviations of the results from eight independent experiments in the biofilm formation assay, and from three experiments in the swimming motility tests. Values that are significantly different based on an unpaired two-tailed *t*-test are indicated by asterisks, as follows: **: *p* < 0.009.

**Figure 4 ijms-23-00159-f004:**
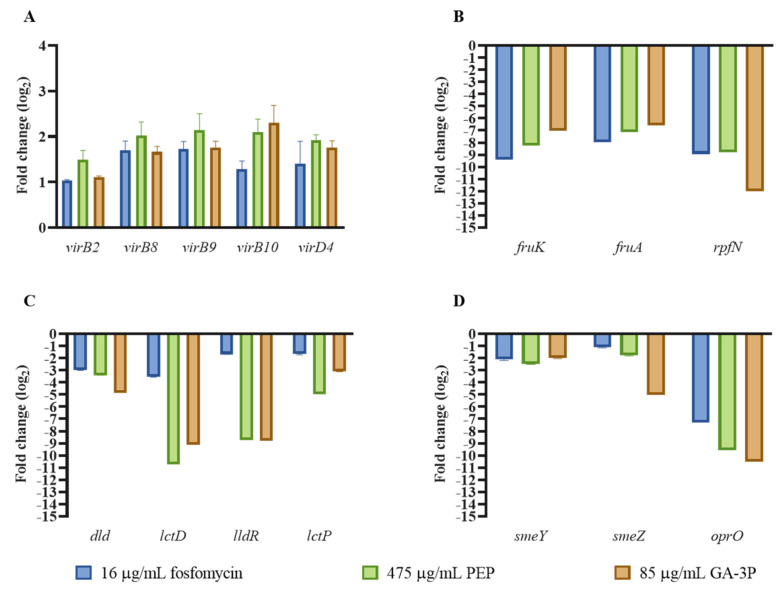
Analysis of expression by real-time PCR in the presence of fosfomycin, PEP, or GA-3P: (**A**) T4SS expression, (**B**) fructose PTS expression, (**C**) lactate operon expression and (**D**) expression of transporters after 1 h of incubation with 16 μg/mL fosfomycin, 475 μg/mL PEP, or 85 μg/mL GA-3P. As shown, the expression of T4SS was induced by the three tested compounds (**A**). After the three treatments, the expression level of fructose and lactate operons, as well as important transporters (**B**–**D**), was reduced by 10-fold as compared with the untreated bacteria. Each represented value is the average of three biological replicates. Statistically significant differences compared to untreated D457 were calculated with a *t*-test for paired samples assuming equal variances with all of them.

**Figure 5 ijms-23-00159-f005:**
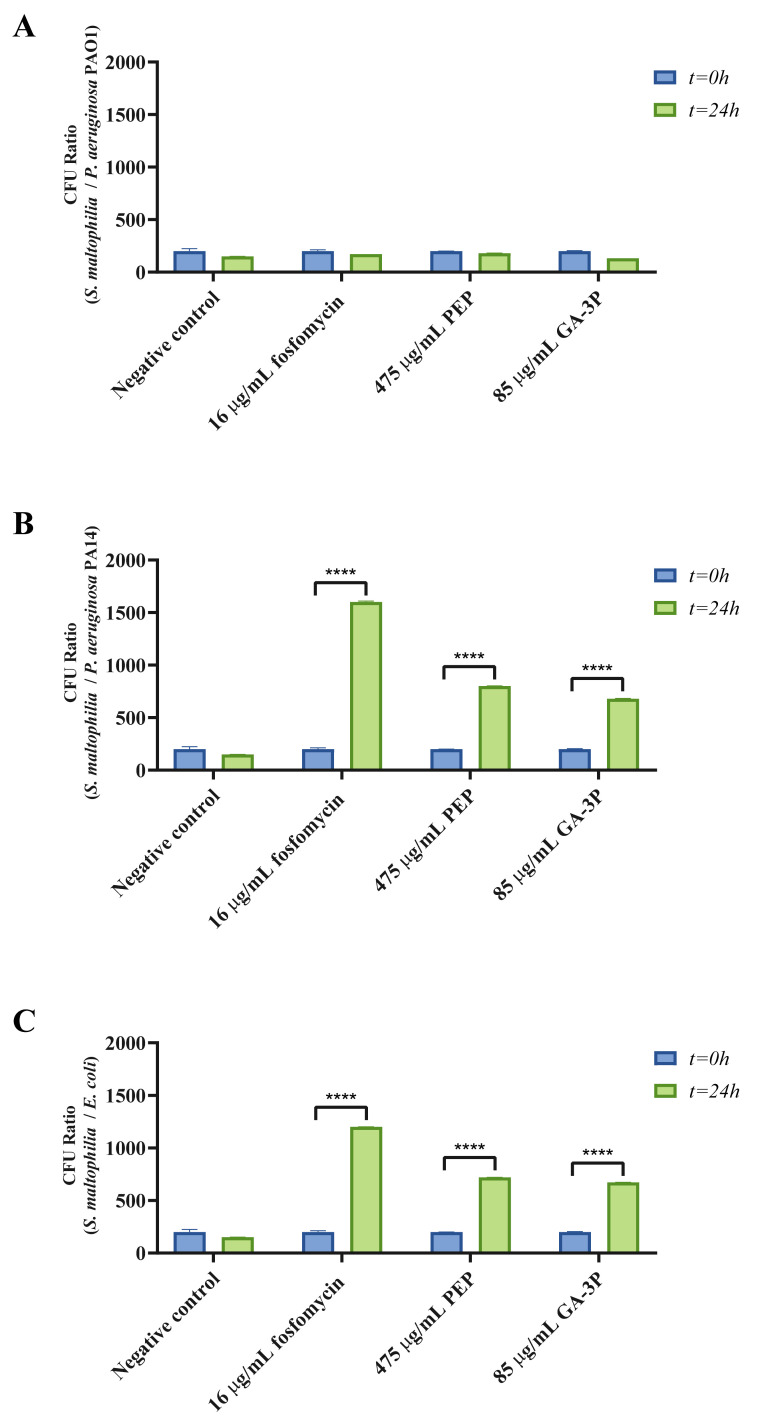
Survival of *S. maltophilia* D457 when co-cultured with heterologous bacteria after fosfomycin, PEP, or GA-3P pretreatments. Results are presented as the ratios of *S. maltophilia* CFUs to the CFUs of other species at t = 0 and t = 24 h. On the one hand, tested compounds were not able to improve *S. maltophilia* D457 growth when co-cultured with (**A**) *P. aeruginosa* PAO1. On the other hand, all tested compounds were able to improve *S. maltophilia* D457 growth when co-cultured with (**B**) *P. aeruginosa* PA14 or (**C**) *E. coli*. Values that are significantly different based on an unpaired two-tailed *t*-test are indicated by asterisks as follows: ****: *p* < 0.0001. Data are presented as the means and standard deviations of results from three independent experiments.

**Figure 6 ijms-23-00159-f006:**
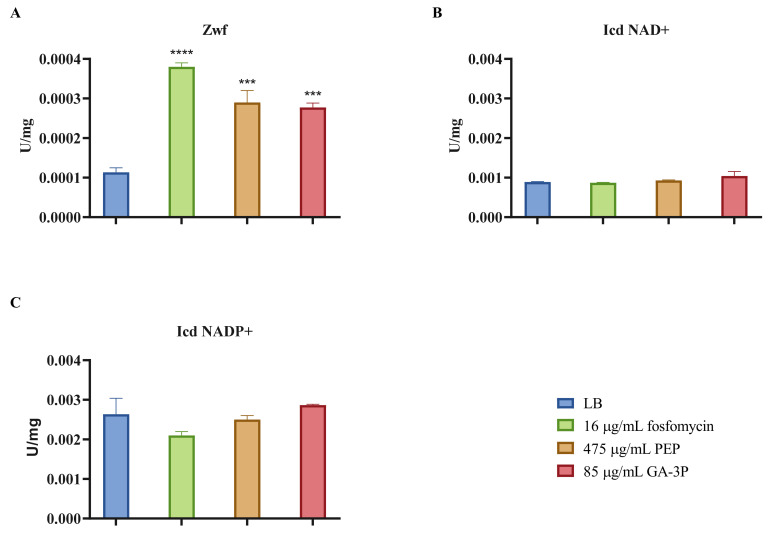
Enzymatic activity of the main dehydrogenases from the D457 strain in the exponential growth phase without treatment, and after 1 h of fosfomycin, PEP, and GA-3P treatment: (**A**) Zwf glucose-6-phosphate dehydrogenase activity; (**B**) Icd (NAD+) isocitrate dehydrogenase NAD+ activity; (**C**) Icd (NADP+) isocitrate dehydrogenase NADP+ activity. Error bars indicate standard deviations for the results from three independent replicates. As shown, the activity of Zwf was higher in treated samples. Statistical significance was calculated by unpaired two-tailed *t*-test; ***: *p* < 0.007; ****: *p* < 0.00001.

**Figure 7 ijms-23-00159-f007:**
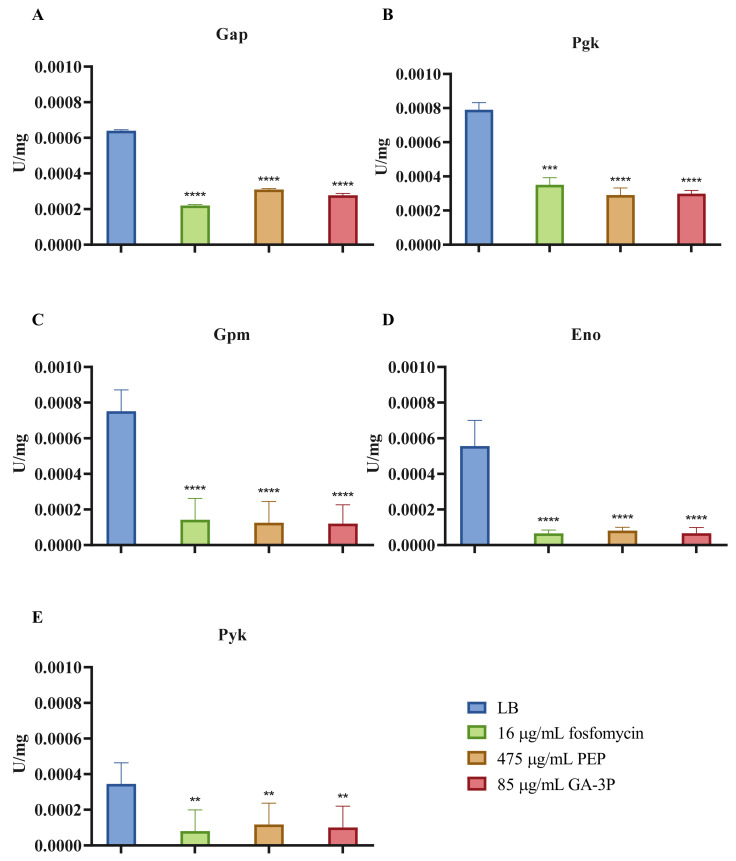
Enzymatic activity of the enzymes from the lower glycolytic pathway of the D457 strain in the exponential growth phase without treatment, or after 1 h of fosfomycin, PEP, or GA-3P treatment: (**A**) Gap glyceraldehyde-3-phosphate dehydrogenase activity; (**B**) Pgk phosphoglycerate kinase activity; (**C**) Gpm phosphoglycerate mutase activity; (**D**) Eno enolase activity; (**E**) Pyk pyruvate kinase activity. Error bars indicate standard deviations for the results from three independent replicates. As shown, both treatments impaired the activity of all of the lower glycolytic enzymes. Values that are significantly different from the values for the samples without treatments were calculated by an unpaired two-tailed *t*-test, and are indicated by asterisks as follows: **: *p* < 0.004; ***: *p* < 0.0007; ****: *p* < 0.0001.

**Figure 8 ijms-23-00159-f008:**
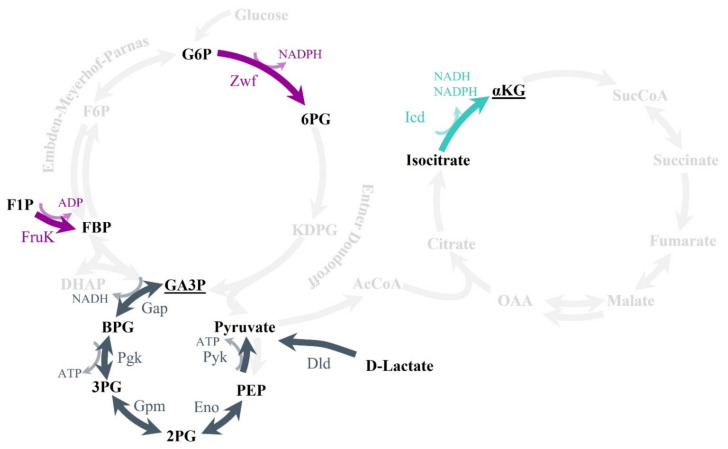
Effects of fosfomycin, PEP, and GA-3P on *S. maltophilia*’s central metabolism. The enzymes whose expression or activity was seen to be modified in the transcriptomics or in the activity measurements are indicated (see text for more details). Substrate abbreviations—G6P: glucose-6-phophate; F6P: fructose-6-phosphate; F1P: fructose-1-phosphate; FBP: fructose-1,6-phosphate; DHAP: dihydroxyacetone-phosphate; GA3P: glyceraldehyde-3-phosphate; BPG: 1,3-bisphosphoglycerate; 3PG: 3-phosphoglycerate; 2PG: 2-phosphoglycerate; PEP: phosphoenolpyruvate; AcCoA: acetyl-coenzyme A; 6PG: 6-phosphogluconate; KDPG: 2-keto-3-deoxy-d-gluconate-6-phosphate; αKG: α-ketoglutarate; SucCoA: succinyl-coenzyme A; OAA: oxaloacetate. Enzymes: Zwf: G6P 1-dehydrogenase; FruK: 1-phosphofructokinase; Gap: GA3P dehydrogenase; Pgk: phosphoglycerate kinase; Gpm: phosphoglycerate mutase; Eno: enolase; Pyk: pyruvate kinase; Dld: D-lactate dehydrogenase; Icd: isocitrate dehydrogenase.

**Table 1 ijms-23-00159-t001:** Number of genes with expression changes in each functional category.

Functional Category	Total Genes	Fosfomycin Transcriptome (Specific)	PEP Transcriptome (Specific)	GA-3P Transcriptome (Specific)
Stress response	55	25 * (0 **)	47 (5)	49 (8)
Chemotaxis and motility	50	28 (9)	39 (2)	34 (2)
Secretion systems	6	0 (0)	6 (1)	5 (0)
Resistance	5	1 (1)	3 (1)	3 (1)
Transport	68	33 (6)	55 (17)	41 (7)
General metabolism	51	11 (7)	36 (31)	40 (30)
Amino acid metabolism	24	14 (2)	19 (1)	18 (3)
Outer membrane	11	2 (1)	9 (2)	8 (1)
Regulators	44	18 (3)	31 (7)	31 (10)
Replication and transcription	21	6 (2)	17 (6)	13 (2)
Translation	47	32 (4)	35 (3)	40 (7)
Iron-related proteins	37	11 (2)	32 (17)	16 (3)
Other genes	138	47 (8)	111 (31)	86 (19)

* Total number within each category and in each transcriptome; ** the number of genes that are specific to each transcriptome is shown in parentheses.

**Table 2 ijms-23-00159-t002:** Fold change (log_2_) of genes related to stress response presenting different levels of expression after the treatments, in comparison with the wild-type D457 strain without treatment.

		Fold Change (log_2_)		
ID—Gene		Fosfomycin	PEP	GA-3P
*SMD_RS00495*		1.17	1.11	1.03
*SMD_RS00575*		0.97	1.21	1.12
*SMD_RS02500*		0.86	1.39	1.18
*SMD_RS02505*		0.67	1.13	1.12
*SMD_RS02645*	*msrB*	1.94	2.48	2.45
*SMD_RS03830*	*msrA*	1.06	1.58	1.62
*SMD_RS04070*		0.42	1.05	1.04
*SMD_RS04515*	*grxC*	0.83	1.05	1.06
*SMD_RS04605*	*lon*	1.16	1.41	1.4
*SMD_RS05310*	*soxR*	1.21	1.59	1.22
*SMD_RS06170*		0.99	1.27	1.32
*SMD_RS06465*		1.21	1.38	1.38
*SMD_RS09215*	*htpG*	1.4	1.75	1.79
*SMD_RS09380*		0.64	0.95	1.16
*SMD_RS09475*	*hrcA*	0.97	1.31	1.36
*SMD_RS09480*	*grpE*	1.19	1.55	1.56
*SMD_RS09485*	*dnaK*	1.08	1.36	1.39
*SMD_RS09490*	*dnaJ*	0.88	1.05	0.99
*SMD_RS11005*		1.46	1.23	1.04
*SMD_RS11055*	*clpA*	0.95	1.35	1.42
*SMD_RS11800*		0.97	1.06	0.89
*SMD_RS13010*		−0.28	−1.07	−1.15
*SMD_RS13025*		−0.02	0.98	1.24
*SMD_RS13030*		−0.18	0.87	1.12
*SMD_RS13045*		1.13	1.47	1.41
*SMD_RS13060*		−1.01	−1.25	−0.77
*SMD_RS14070*		1.35	1.59	1.52
*SMD_RS14255*	*sufA*	0.79	1.02	1.01
*SMD_RS14660*	*sodA*	1.2	1.59	1.64
*SMD_RS15050*	*htpX*	1.01	1.08	1.22
*SMD_RS15740*		0.8	1.19	1.01
*SMD_RS15845*	*tldD*	0.8	0.99	1
*SMD_RS15930*	*trxA*	1.71	1.98	1.91
*SMD_RS16120*	*sphB*	0.87	1.36	1.12
*SMD_RS16195*		1.23	1.52	1.48
*SMD_RS16245*		1.24	1.41	1.56
*SMD_RS16285*	*cstA*	0.61	0.25	1.12
*SMD_RS16290*		0.37	0.18	1.03
*SMD_RS16330*		0.92	1.09	0.91
*SMD_RS16640*	*hslO*	1.35	1.52	1.6
*SMD_RS17335*	*clpB*	1.24	1.74	1.77
*SMD_RS17345*		0.74	1.13	1.42
*SMD_RS18465*		1.8	2.12	1.85
*SMD_RS18955*	*hslU*	1.79	2.14	2.2
*SMD_RS18960*	*hslV*	1.71	2.19	2.2
*SMD_RS18990*	*ptrB*	1.08	1.46	1.42
*SMD_RS19000*		0.91	1.02	0.88
*SMD_RS19045*	*prlC*	1.39	1.78	1.77
*SMD_RS19280*		0.67	1.12	1.16
*SMD_RS19325*		0.9	1.48	0.94
*SMD_RS19815*	*groL*	0.76	0.99	1.05
*SMD_RS19820*	*groES*	1.08	1.41	1.39
*SMD_RS21040*	*gst6*	0.77	1.18	1.19
*SMD_RS21540*		1.09	1.32	1.73
*SMD_RS18790*		0.62	0.64	1.12

**Table 3 ijms-23-00159-t003:** Fold change (log_2_) of T4SS genes presenting different levels of expression after the treatments, in comparison with the wild-type D457 strain without treatment.

		Fold Change (log_2_)		
ID—Gene		Fosfomycin	PEP	GA-3P
*SMD_RS13770*	*virB2*	0.83	1.17	1.06
*SMD_RS13785*	*virB10*	0.94	1.3	1.11
*SMD_RS13790*	*virB9*	0.98	1.4	1.24
*SMD_RS13795*	*virB8*	0.92	1.23	1.09
*SMD_RS13805*	*virD4*	0.64	1.16	1.03

**Table 4 ijms-23-00159-t004:** Main metabolic and transport genes presenting changes in their expression levels in the presence of fosfomycin, PEP, or GA-3P.

		Fold Change (log_2_)		
ID—Gene or Product		Fosfomycin	PEP	GA-3P
*SMD_RS11695*	*ptsP*	−4.26	−2.63	−1.6
*SMD_RS11700*	*fruK*	−5.08	−5.11	−1.76
*SMD_RS11705*	*fruA*	−5.06	−5.09	−1.78
*SMD_RS11710*	*rpfN*	−3.53	−4.93	−2.05
*SMD_RS13320*	*dld*	−1.53	−3.25	−0.74
*SMD_RS13325*	*lctD*	−1.4	−2.9	−0.67
*SMD_RS13330*	*lldR*	−1.42	−2.3	−0.77
*SMD_RS13335*	*lctP*	−1.34	−3.77	−0.56
*SMD_RS14045*	*hutF*	1.11	1.29	1
*SMD_RS14050*	*hutI*	1.08	1.59	1.33
*SMD_RS14055*	*hutH*	1.3	1.53	1.45
*SMD_RS14060*	*hutG*	1.38	1.85	1.81
*SMD_RS14065*	*hutU*	2.2	2.51	2.52

**Table 5 ijms-23-00159-t005:** Expression of the genes encoding the main *S. maltophilia* dehydrogenase enzymes and the lower glycolysis enzymes in the presence of fosfomycin, PEP, or GA-3P.

		Fold Change (log_2_)		
ID—Gene		Fosfomycin	PEP	GA-3P
*SMD_RS09135*	*zwf*	−0.21	−0.02	0.15
*SMD_RS04525*	*icd* (NAD+)	0.24	0.15	0.17
*SMD_RS20085*	*icd* (NADP+)	0.56	0.47	0.38
*SMD_RS17680*	*gap*	0.24	0.35	0.36
*SMD_RS17665*	*pgk*	0.48	0.4	0.31
*SMD_RS06650*	*gpmA*	0.26	0.33	0.39
*SMD_RS04430*	*gpm*	−0.53	−1.11	−1.21
*SMD_RS08765*	*eno*	0.64	0.69	0.6
*SMD_RS17655*	*pyk*	0.22	0.43	0.14

**Table 6 ijms-23-00159-t006:** Fold change (log_2_) of efflux pumps and porins presenting different levels of expression after the treatments, in comparison with the wild-type D457 strain without treatment.

		Fold Change (log_2_)		
ID—Gene		Fosfomycin	PEP	GA-3P
SMD_RS10440	*smeY*	−0.99	−1.77	−1.76
SMD_RS10445	*smeZ*	−1.09	−1.78	−1.65
SMD_RS18400	*oprO*	−6.52	−6.55	−2.8

**Table 7 ijms-23-00159-t007:** Effects of the SmeYZ-inhibitory compounds on the susceptibility of *S. maltophilia* to antibiotics.

	MIC (μg/mL)			
	LB	Fosfomycin (16 μg/mL)	PEP (475 μg/mL)	GA-3P (85 μg/mL)
Amikacin	6	3	3	3
Tobramycin	4	2	1.5	2
Gentamicin	2	1.25	1	1.25
Tetracycline	2	1.5	0.75	1
SXT	0.3	0.19	0.125	0.19

## Data Availability

The RNA-Seq data reported in this publication have been deposited in NCBI’s Gene Expression Omnibus [77], and are accessible via GEO Series accession number GSE181918. All other information is included in the manuscript.

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
