# Peer review of "The Antibiotic Fosfomycin Mimics the Effects of the Intermediate Metabolites Phosphoenolpyruvate and Glyceraldehyde-3-Phosphate on the Stenotrophomonas maltophilia Transcriptome"

_ijms, 2021, doi:10.3390/ijms23010159_

Round 1

Reviewer 1 Report

In the present work, Gil-Gil et al perform RNAseq analyses on Stenotrophomonas maltophilia cells to explore stress-induced changes to S. maltophilia gene expression caused by fosfomycin, phosphoenolpyruvate (PEP) or glyceraldehyde-3-phosphate (G3P). They find that transcriptomes induced by these three agents are very similar, due to shared homology between fosfomycin and PEP and shared metabolic pathway activity between PEP and G3P. They perform validation experiments revealing shared phenotypes at the cellular, multicellular, inter-cellular, genetic and metabolic levels. Overall, this is a nice study and I only have two minor suggestions for the authors:

  1. it would be helpful if early on (e.g., Fig 1) there could be a cartoon visualization of the EMP pathway to help orient the reader to the genes measured and manipulated in the validation experiments
  2. They grey shading and text in Fig. 8 is too light for me to read when printed

Author Response

We appreciate the positive opinion of the referee concerning our world. Regarding que queries:

  1. A cartoon of the metabolic pathway affected in the fosfomycin resistant mutants has been included within Figure 1
  2. The Figure has been modified

Reviewer 2 Report

In the present study “The Antibiotic Fosfomycin Mimics the Effect of the Intermediate Metabolites Phosphoenolpyruvate and Glyceraldehyde-3-phosphate on Stenotrophomonas maltophilia Transcriptome” authors have done impressive work to elucidate the effect of Fosfomycin, PEP, and GA-3P on the bacterial transcriptome. A wide range of assays and genome-based techniques has been used to justify the hypothesis. The Paper is written in a clear and informative manner and also supported by statistical analysis.

I have read the entire paper and think that some minor revisions are required to go further with the manuscript.

Some Minor questions and Suggestions are-

1- The quality of figure-1 needs to be improved.

2- Is it predetermined that MIC of PEP and GA-3P is 1900 and 170 against S. maltophilia?

Why do authors select only 3 specifically selected ranges of concentrations in every case?  (Figure-1)

3- Table-2 needs to be formatted, I think the column has been misplaced. Please correct it.

4- The fold changes in gene expression level in the last two columns (I think for PEP and GA-3P) are almost similar in most of the cases. What could be the possible reason? (Table-2)

5- Authors said that there is no changes have been observed in swimming motility on the addition of any of the compounds, but as far as the graph is concerned there is a significant change in the case of GA-3P, It seems decreased by almost 0.5 units. (Figure-3)

Kindly Explain

6- Kindly make appropriate corrections in table formatting. Headings has been misplaced.

(Table-3, 4, 5, 6)

7- Authors have performed and discussed the expressions in untreated bacteria, but they did not show them in the graph. It would be very impressive if they add the case of untreated bacteria also. (Figure-4)

9- From where the strains were obtained? (Line- 615)

10- What were the criteria for the selection of a specific concentration range of the different compounds? (Line- 622-623)

11- Did the authors check the activity at other time duration also? What was the reason to select 60 min for the reaction in the case of all the three compounds? (Line 662-663)

Author Response

We appreciate the positive opinion of the referee concerning our work. Regarding referee's queries

1- The quality of figure-1 needs to be improved.

Answer: Done

2- Is it predetermined that MIC of PEP and GA-3P is 1900 and 170 against S. maltophiliaWhy do authors select only 3 specifically selected ranges of concentrations in every case?  (Figure-1)

Answer: MIC values have been determined according to double dilution MIC assays. Concentration ranges were chosen for being subinhibitory concentrations from a concentration showing a fitness costs to a lower concentration with minimal fitness cost.

3- Table-2 needs to be formatted, I think the column has been misplaced. Please correct it.

Answer: Done. All tables have been formatted.

4- The fold changes in gene expression level in the last two columns (I think for PEP and GA-3P) are almost similar in most of the cases. What could be the possible reason? (Table-2)

Answer: We agree this is notable; but these are the results we got and the reasons behind the similarities remain speculative. It would be possible that the similarity of both transcriptomes is due to the fact that both compounds are metabolites of the same metabolic pathway, lower glycolysis. This fact would explain the fact that both metabolites cause the same effects on bacterial physiology.

5- Authors said that there is no changes have been observed in swimming motility on the addition of any of the compounds, but as far as the graph is concerned there is a significant change in the case of GA-3P, It seems decreased by almost 0.5 units. (Figure-3)

Kindly Explain

Answer: It is true that according to the images, it may seem that there are differences, but sn unpaired two-tail t test was carried out and these putative differences were not statistically significant. This is better discusses now.

6- Kindly make appropriate corrections in table formatting. Headings has been misplaced.

(Table-3, 4, 5, 6)

Answer: Done

7- Authors have performed and discussed the expressions in untreated bacteria, but they did not show them in the graph. It would be very impressive if they add the case of untreated bacteria also. (Figure-4)

Answer. We appreciate the comment, but we did not include the absolute values. In all cases expression changes are showed as the fold change respect to untreated bacteria; which were considered as controla

9- From where the strains were obtained? (Line- 615)

Answer: S. maltophilia D457 is a model strain, used in several works, which was originally obtained from a bronchial aspirate.The appropriate reference is now included.

10- What were the criteria for the selection of a specific concentration range of the different compounds? (Line- 622-623)

Answer: As stated above, concentrations were chosen as those that were subinhibitory and did not impose a high effect on fitness.

11- Did the authors check the activity at other time duration also? What was the reason to select 60 min for the reaction in the case of all the three compounds? (Line 662-663)

Answer: We only checked the activity after 60 min. We choose the same time in all cases for comparing the effects among treatments at this time.